

# A simple model of ozone-temperature coupling in the tropical lower stratosphere

William J. Randel[1,2], Fei Wu[1], Alison Ming[3], Peter Hitchcock[4]

[1]National Center for Atmospheric Research, Boulder, CO, USA

[2]COSMIC Program, University Corporation for Atmospheric Research, Boulder, CO, USA

[3]University of Cambridge, Cambridge, UK

[4]Cornell University, Ithaca, NY

*Correspondence to*: William Randel (randel@ucar.edu)

**Abstract.** Observations show strong correlations between large-scale ozone and temperature variations in the tropical lower stratosphere across a wide range of time scales. We quantify this behavior using monthly records of ozone and temperature data from SHADOZ tropical balloon measurements (1998-2016), along with global satellite data from Aura MLS and GPS radio occultation over 2004-2018. The observational data demonstrate strong in-phase ozone-temperature coherence

spanning sub-seasonal, annual and interannual time scales, and the slope of the ozone-temperature relationship (O3/T) varies as a function of time scale and altitude. We compare the observations to idealized calculations based on the coupled zonal mean thermodynamic and ozone continuity equations, including ozone radiative feedbacks on temperature, where both temperature and ozone respond in a coupled manner to variations in the tropical upwelling Brewer-Dobson circulation. These calculations can approximately explain the observed (O3/T) amplitude and phase relationships, including sensitivity to

time scale and altitude, and highlight distinct balances for 'fast' variations (periods < 150 days, controlled by transport across background vertical gradients) and 'slow' coupling (seasonal and interannual variations, controlled by radiative balances).

## 1 Introduction

Large-scale ozone and temperature variations in the tropical lower stratosphere exhibit strong correlations across a range of time scales. This behavior is well-known for the annual cycle in the lower stratosphere (Chae and Sherwood, 2007; Randel et al, 2007) and for interannual variations linked to the quasi-biennial oscillation (QBO) (e.g. Hasebe et al, 1994;

Baldwin et al, 2001; Witte et al, 2008; Hauchecorne et al, 2010) and El Nino Southern Oscillation (ENSO; Randel et al, 2009). Abalos et al (2012; 2013) and Gilford et al (2016) also note strong ozone-temperature correlations in this region across a range of time scales. Calculations have shown that the radiative effects of ozone feed back onto and enhance temperature variations, and this topic has been well-studied as related to the annual cycle in the tropical lower stratosphere





(Chae and Sherwood, 2007; Fueglistaler et al, 2011; Ming et al, 2017; Gilford and Solomon, 2017), and also by Forster et al (2007) and Polvani and Solomon (2012) for decadal-scale trends. Yook et al (2020) showed that ozone feedback is an important contribution to tropical stratospheric thermal variability in global models. Birner and Charlesworth (2017) and Dacie et al (2019) have demonstrated strong sensitivity of tropical stratospheric temperatures to ozone using idealized one-dimensional model calculations, following the earlier results of Thuburn and Craig (2002). Charlesworth et al (2019)

extended that work to study transient ozone-temperature feedbacks, highlighting larger effects for low frequency variations (periods longer than about half a year).

The dominant mechanism for strong ozone-temperature correlations in the tropical lower stratosphere is relatively simple, namely, variations in upwelling (i.e. fluctuations in the tropical Brewer-Dobson circulation) acting on the strong background vertical gradients of both ozone and potential temperature, leading to correlated variability. This behavior was

quantified from observations and model simulations in Abalos et al (2012; 2013), highlighting the control of upwelling for forcing transient variations in temperature, ozone and other trace species with strong vertical gradients, such as carbon monoxide (CO). The radiative feedback of ozone to temperature imparts further complexity to this simple system, and that is the focus of this work. Here we update the observational evidence of ozone-temperature coupling based on long records of tropical balloon measurements from SHADOZ (Thompson et al, 2003), focusing on annual and interannual variability. We

also analyze over a decade of continuous satellite measurements to quantify ozone-temperature coherence and phase in the tropical stratosphere over a continuous range of time scales. We compare the observational results with calculations based on the coupled zonal mean thermodynamic and ozone continuity equations, simplified to approximate the balances in the tropical lower stratosphere, and including ozone feedback on temperature. Our goal is to explain the salient features of ozone-temperature (O3-T) coupling from observations in a relatively simple framework, including the frequency and altitude

dependences of the (O3/T) amplitude and phase relationships. These results are a complement to the recent analyses of Birner and Charlesworth (2017) and Charlesworth et al (2019), based on a very different model.

## 2 Data and Analyses

### 2.1 SHADOZ ozone and temperature

The Southern Hemisphere Additional Ozonesonde (SHADOZ) network consists of ~12 stations covering a range of

longitudes over the latitude band ~ 10º N – 20º S, with measurements beginning in 1998 (Thompson et al, 2003). Recent reprocessing of the data is discussed in Witte et al (2017) and Thompson et al (2017). The SHADOZ balloons measure ozone and pressure-temperature-wind profiles, with effective vertical resolution of ~50-100m. The data used here are sampled with 0.5 km vertical spacing, and we focus on altitudes 15-30 km. We analyze data from SHADOZ stations with long and continuous records, updated from Randel and Thompson (2011). There are typically 2-4 observations per month at

each of the SHADOZ stations, which we combine into simple monthly averages. The stratospheric segment of the ozone profile exhibits a high degree of longitudinal symmetry (Thompson et al, 2003; Randel et al, 2007; Randel and Thompson,



2011) and we combine monthly average results from all stations to provide approximate zonal average monthly means of ozone and temperature, with data covering 1998-2016.

### 2.2 Aura MLS ozone and GPS temperature

Satellite ozone measurements from the Aura Microwave Limb Sounder (MLS) are analyzed for the period September 2004 – May 2018. We use retrieval version 4.2 (Livesey et al, 2018). Data are available for standard pressure levels (12 per decade) covering 316 hPa to above 1 hPa; the vertical resolution of the grid is ~1.3 km, but the resolution of the MLS measurements is closer to ~3 km (i.e. the data are oversampled). Data quality for MLS v4.2 ozone is discussed in Livesey et al (2018). Our analyses focus on the latitude band 10º N-S, and we calculate zonal mean values for 5-day (pentad)

averages. Some isolated data gaps are filled by linear interpolation in time. This provides a long, continuous time series of MLS ozone covering 998 pentads (4990 days).

Temperature data are obtained from GPS radio occultation, which provides high quality and high vertical resolution (~1 km) measurements over 10-30 km, and near-global sampling (Anthes et al, 2008). We combine measurements from several different GPS satellites for the period overlapping the MLS ozone data (September 2004 – May 2018), and construct

pentad time series from data over 10º N-S to match the MLS ozone time series discussed above. We focus on altitude levels close to the MLS ozone grid. The time series analyzed here are an update of the data analyzed in Randel and Wu (2015), and further details are discussed there.

### 2.3 Spectrum analysis

We include spectrum and cross-spectrum analysis of the satellite-derived ozone and temperature time series to

quantify frequency-dependent relationships. Spectra are calculated by direct Fourier transform of the 998-pentad time series for both ozone and temperature, resolving periods of 4990 to 10 days, with a frequency resolution of $\Delta\omega = (2\pi/4990 \text{ days})$. Calculations are based on standard formulas in Jenkins and Watts (1968). Power spectra are smoothed with a Gaussian-shaped smoothing window with half-width $2\Delta\omega$. Ozone-temperature amplitude ratios, coherence squared and phase spectra are calculated using a wider bandwidth $(10\Delta\omega)$ to enhance statistical stability. This results in approximately 10 independent

Fourier harmonics for each spectral estimate, and the resulting 95% significance level for the coh2 statistic is 0.45. The high- and low-frequency ends of the spectra are smoothed using one-sided Gaussian smoothing so that significance levels are somewhat higher.





### 3 Simplified zonal mean theory

#### 3.1 Coupled thermodynamic and ozone continuity equations

90       We explore the coupling of ozone and temperature based on the zonal mean thermodynamic and ozone continuity equations, simplified to approximate behavior in the tropical lower stratosphere, namely neglecting mean meridional advection and eddy forcing terms. The zonal mean thermodynamic equation in transformed Eulerian-mean coordinates, using a log-pressure vertical coordinate (Andrews et al, 1987) is:

$$\partial T/\partial t = -v^*(\partial T/\partial y) - w^* S + \text{eddy terms} + Q \qquad (1)$$

Here T is zonally averaged temperature, $(v^*, w^*)$ are components of the residual meridional circulation, S is a stability parameter, and Q is the zonal mean diabatic heating rate. In the tropical lower stratosphere the $v^*$ and eddy forcing terms are relatively small (Abalos et al, 2013), so that the approximate thermodynamic balance is:

$$\partial T/\partial t = -w^* S + Q \qquad (2)$$

In this work we specify the zonal mean diabatic forcing Q with two components $Q = Q_{relaxation} + Q_{ozone}$, representing radiative
relaxation and ozone forcing of temperature, respectively. We assume radiative relaxation is proportional to temperature, $Q_{relaxation} = -\alpha (T-T_{eq})$, with $T_{eq}$ an equilibrium temperature and $\alpha$ an inverse radiative damping time scale (Andrews et al, 1987; Hitchcock et al, 2010). $\alpha$ is obtained from the results of Hitchcock et al (2010) as discussed below. In addition, correlated variations in ozone produce a positive radiative feedback on temperature (e.g. Fueglistaler et al, 2011; Gilford et al, 2017; Ming et al, 2017), and while this is in general a non-local effect (in altitude), for simplicity we model the
temperature tendency as proportional to the local ozone anomaly: $Q_{ozone} = \beta (X - X_{eq})$. Here $X$ is zonal mean ozone mixing ratio, $X_{eq}$ is a background equilibrium ozone value and $\beta$ is a constant derived from radiative transfer calculations (described below). Based on these simplified assumptions, the zonal mean thermodynamic equation becomes:

$$\partial T/\partial t = -w^* S - \alpha (T-T_{eq}) + \beta (X-X_{eq}) \qquad (3)$$

Assuming harmonic time expansions of the form $T(t) = \sum T_\sigma \exp^{i\sigma t}$, with $\sigma$ the angular frequency ($2\pi/\text{period}$), and likewise
for $w^*(t)$ and $X(t)$, and assuming $T_{eq}$, $X_{eq}$ and S are constant in time, Eq. 3 can be rewritten as an equation for each harmonic component:

$$i\sigma T_\sigma = -w^*_\sigma S - \alpha T_\sigma + \beta X_\sigma \qquad (4)$$

A similar analysis is applied to the zonal mean ozone continuity equation (Andrews et al, 1987, Eq. 9.4.13):

$$\partial X/\partial t = -v^*(\partial X/\partial y) - w^*(\partial X/\partial z) + \text{eddy terms} + P - L \qquad (5)$$

Here P-L represents chemical ozone production minus loss terms. In contrast to the thermodynamic balance discussed above, the eddy terms for ozone transport in the tropical lower stratosphere are not negligible, and there is a maximum during boreal summer near the tropopause related to transport from the subtropical monsoon circulations (Konopka et al, 2009, 2010; Abalos et al, 2013). However, for simplicity in our idealized calculations the eddy terms are neglected here, along with the $v^*$ term. This yields:





$$\partial X/\partial t = -w^* X_z + P - L, \qquad (6)$$

where we have defined $X_z = (\partial X/\partial z)$. In the tropical lower stratosphere ozone production minus loss is positive and is relatively constant in time, with a small semi-annual variation in production following solar inclination (Abalos et al, 2013). We parameterize ozone loss as $L = -\delta (X - X_{eq})$, where $\delta$ is the inverse lifetime of ozone, obtained from model calculations as described below. Assuming a constant production rate (P) and a constant background ozone gradient $X_z$, the harmonic

expansion of Eq. 6 is then given by the simple balance:

$$i\sigma X_\sigma = -w^*_\sigma X_z - \delta X_\sigma \qquad (7)$$

We show idealized model calculations below including realistic ozone damping estimates (along with results for no damping), which demonstrate that ozone damping has a relatively small influence for the majority of results.

The balances in the simplified equations (Eq. 4 and 7) are driven by temperature and ozone responses to imposed

vertical velocity variations (w*) in the tropical lower stratosphere, as is observed and derived from model simulations (Abalos et al, 2012; 2013). Temperature is furthermore influenced by radiative damping ($\alpha$ term) and the radiative response to ozone changes ($\beta$ term), while ozone balance includes damping ($\delta$ term). Equations 4 and 7 can be combined to eliminate the w* dependence to obtain a single equation relating ozone and temperature harmonic components, in particular the ozone/temperature ratio as a function of frequency:

$$(X_\sigma/T_\sigma) = (X_z/S)\,(\alpha + i\sigma)/(\beta' + \delta + i\sigma), \qquad (8)$$

with $\beta' = (X_z/S)\,\beta$. This can be rewritten as:

$$(X_\sigma/T_\sigma) = A + iB$$
$$\text{with } A = (X_z/S)\,(\sigma^2 + \alpha(\beta' + \delta))/(\sigma^2 + (\beta' + \delta)^2)$$
$$\text{and } B = (X_z/S)\,\sigma(\beta' + \delta - \alpha)/(\sigma^2 + (\beta' + \delta)^2)$$

Here $(X_z/S)$ is a key parameter related to the ratio of ozone vertical gradient to background stability (potential temperature gradient), which is derived directly from the time average temperature and ozone profile data; the vertical profile of $(X_z/S)$ is shown in Fig. 1c. We note some small (~10%) seasonal and interannual variations to the individual $X_z$ and S terms in the tropical lower stratosphere, but these follow each other and the ratio $(X_z/S)$ is more nearly constant. Equation 8 can be rewritten as expressions for $(X_\sigma/T_\sigma)$ amplitude and phase:

$$(X_\sigma/T_\sigma)_{amplitude} = \text{sqrt}\,(A^2 + B^2) \qquad (9a)$$
$$(X_\sigma/T_\sigma)_{phase} = \tan^{-1}(B/A) \qquad (9b)$$

Our analyses focus on evaluating the quantity $(X_\sigma/T_\sigma)$ as a metric for ozone-temperature coupling, and below we test results from this idealized model with $(X_\sigma/T_\sigma)$ amplitude and phase derived from observations. The observational data are based on measurements from SHADOZ and MLS/GPS in the deep tropics over ~10° N-S, and hence represent this tropical average.

We note that Stolarski et al (2014) and Tweedy et al (2017) highlight distinct ozone behavior in southern tropics vs. northern





tropics in the region up to ~18 km, due to influence of the boreal summer monsoons. This could potentially impact our comparisons close to the tropopause, but should have less influence above ~18 km.

The $(X_\sigma/T_\sigma)$ ratio in Eq. 8 is a generally complex function of $\sigma$, $\alpha$, $\beta'$, $\delta$ and $(X_z/S)$, but it is useful to consider the high- and low-frequency limits (compared to the inverse time scales $\alpha$, $\beta'$ and $\delta$). For high frequencies ($\sigma \gg \alpha$, $\beta'$, $\delta$), the

$(X_\sigma/T_\sigma)$ ratio simplifies to $\sim (X_z/S)$, i.e. the ozone and temperature anomalies are in phase, with a ratio simply related to the background gradients in ozone and potential temperature. For the low frequency limit ($\sigma \sim 0$), $(X_\sigma/T_\sigma) \sim (X_z/S)$ $(\alpha/(\beta'+\delta))$, which simplifies to $(\alpha/\beta)$ for small $\delta$. This expresses an in-phase balance of ozone and temperature associated with the $\alpha$ and $\beta$ radiative terms in the thermodynamic equation (Eq. 3), i.e. heating from ozone anomalies balances radiative cooling.

### 160 3.2 Estimating $\alpha$, $\beta$ and $\delta$ from model calculations

Our calculations use a vertical profile of $\alpha$ in the tropical stratosphere derived by Hitchcock et al (2010), as shown in Fig. 1a. These results are based on regressions derived from radiative heating rates and temperatures output from a chemistry-climate model. We note that there are several uncertainties inherent in these calculations, including factors such as tropospheric clouds influencing lower stratospheric heating rates and dependence on the vertical scale of temperature

perturbations (Hartmann et al, 2001; Hitchcock et al, 2010). The overall structure and magnitude of $\alpha$ used here is consistent with other published estimates, e.g. Newman and Rosenfield (1997) and Randel and Wu (2015).

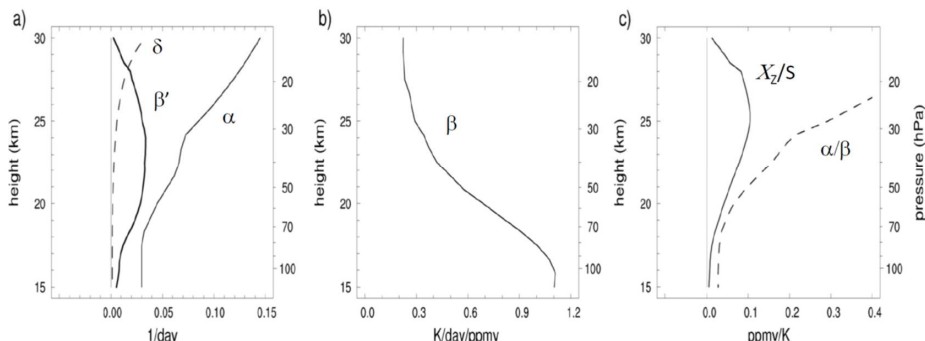

Figure 1. Vertical profiles of parameters used in the theoretical model calculations. (a) $\alpha$, $\beta'$ and $\delta$, b) $\beta$ and c) $(X_z/S)$ and $(\alpha/\beta)$

We estimate vertical profiles of the parameter $\beta$ from radiative transfer calculations using a modified version of the Morcrette (1991) radiation scheme (Zhong and Haigh, 1995). The calculations use realistic background temperature, ozone and water vapor profiles, and carbon dioxide is assumed to be well mixed with a volume mixing ratio of 360 ppmv. Shortwave heating rates are calculated as diurnal averages, including realistic surface albedo, and all calculations assume





clear-sky conditions. β is derived by applying a 0.1 ppmv perturbation to the ozone field at each vertical level, and
calculating the ratio of the instantaneous heating rate change at that level to the amplitude of the ozone perturbation. The
resulting profile of β is shown in Fig. 1b, with typical values of 0.3-1.0 (K/day/ppmv), decreasing in altitude away from the
tropopause. The vertical structure of β' = $(X_z/S)$ β is included in Fig. 1a, showing a magnitude somewhat smaller than α
throughout the profile. This in turn implies a negative $(X_\sigma/T_\sigma)_{phase}$ from Eqns. 8-9b (including a realistic small δ), i.e. ozone
leads temperature in the coupled response based on these parameters, although as shown below the phase difference turns
out to be small. Vertical profile of the quantity (α/β) (zero frequency limit for $(X_\sigma/T_\sigma)$, for small δ) is included in Fig. 1c,
showing increase from the tropopause to the middle stratosphere with values substantially larger than $(X_z/S)$.

         We derived an estimate of the damping rate δ(z) for ozone from simulations of the Whole Atmosphere Community
Climate Model (WACCM; Marsh et al, 2013), which includes a sophisticated stratospheric ozone chemical scheme. These
calculations use daily zonal average output of ozone amount (X) and photochemical ozone loss rate (L) as a function of
latitude and altitude, and we take an annual average of their ratio: δ(z) = (L/X), averaging results over 10° N-S.  The
resulting vertical profile of δ is shown in Fig. 1a, showing very small damping (long ozone lifetimes) in the lower
stratosphere, increasing to slightly larger values in the middle stratosphere (damping time scale of ~30 days at 30 km).
Calculations below show idealized model results including these realistic values of δ, and for comparison we also include
results for δ=0. Including realistic values of ozone damping has almost no influence on calculations in the lower stratosphere
because of the very small damping. Damping can have a small but noticeable effect at higher altitudes for lower frequency
variations (reducing the $X_\sigma/T_\sigma$ ratios), but still only accounts for ~10% effect.

## 4 Ozone and temperature observations

### 4.1 Annual and QBO variability in SHADOZ ozone and temperature

         The approximately monthly sampling of SHADOZ data allows characterization of the annual cycle and interannual
variations of tropical stratospheric ozone and temperature. There is a relatively large annual cycle in ozone and temperature
in the tropical lower stratosphere over ~16-22 km, with relative maxima during boreal summer and ozone having a slight
phase delay compared to temperature. Figure 2a shows this behavior for the 18 km level, near the peak of the annual cycle.
This correlated ozone-temperature behavior is mainly a response to the annual cycle in tropical upwelling (Randel et al,
2007); horizontal transport from the boreal summer monsoons also contributes to the seasonal maximum in ozone close to
the tropopause (Konopka et al, 2009, 2010; Stolarski et al, 2014; Tweedy et al, 2017), but mean upwelling is the dominant
mechanism at and above 19 km (Abalos et al, 2013). Figure 2b shows the corresponding seasonal cycles at 24 km,
highlighting mirror-image variations in ozone and temperature with a stronger semi-annual variation than that at lower
altitudes (Fig. 2a), with ozone slightly leading temperature. Above 24 km the dominant seasonal variation is semi-annual in





both ozone and temperature. We note that the seasonal variations in Figs. 2a-b are very similar based on the MLS ozone and

GPS temperature data (not shown).

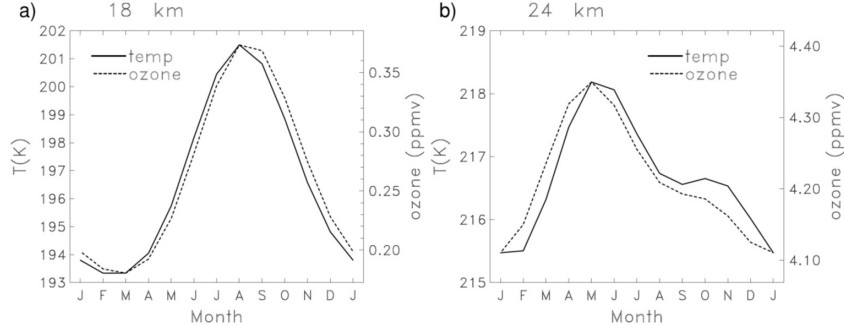

Figure 2. Climatological annual cycles of ozone and temperature at 18 km and 24 km derived from SHADOZ measurements over 1998-2016.

210            Interannual anomalies in ozone and temperature from SHADOZ data over 1998-2016 are shown in Fig. 3, derived by simply subtracting the mean annual cycle. In Fig. 3 ozone anomalies are shown in terms of ozone density (DU/km) instead of mixing ratio, in order to emphasize variations throughout the lower stratosphere. As is well known, there are strong downward propagating anomalies in ozone and temperature linked to the QBO; the ozone and temperature anomalies are approximately in phase over ~17-27 km, and the variations in ozone are small above 27 km due to a transition from

dynamical control in the lower stratosphere to photochemical control above ~27 km (e.g. Chipperfield and Gray, 1992; Park et al, 2017). Episodic ENSO events also result in correlated ozone-temperature variations in the tropical lower stratosphere, for levels from the tropopause to ~22 km (Randel et al, 2009; Calvo et al, 2010). The constructive interference of QBO and ENSO effects can result in large anomalies near and above the tropopause (e.g. Diallo et al, 2018), as seen for the SHADOZ data in 1999-2000 and 2015-2016.




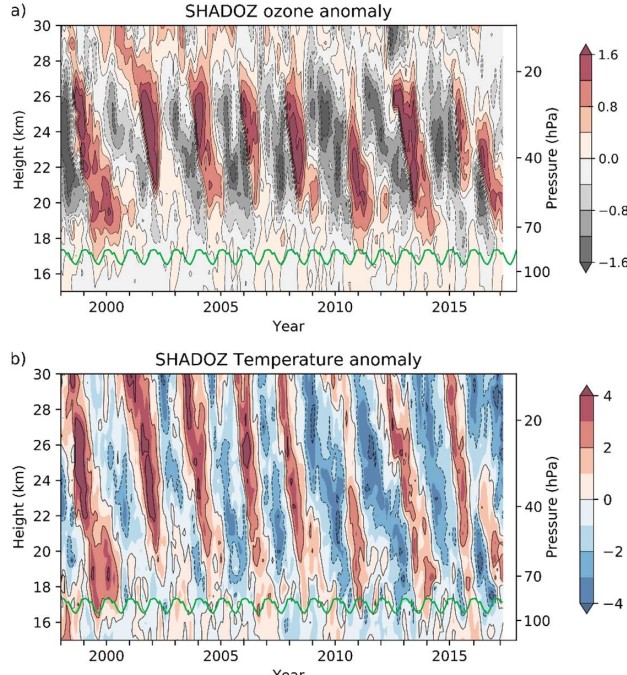


Figure 3. Height-time sections of deseasonalized anomalies in (a) ozone and (b) temperature (K) from SHADOZ data. Ozone anomalies are expressed in terms of ozone density (DU/km) to emphasize variability throughout the lower stratosphere. The green lines denote the cold point tropopause.

Figure 4 shows the ozone-temperature correlation derived from the deseasonalized SHADOZ data (from Fig. 3) as a function of altitude and time lag. Strong positive correlations (>0.8) are found over 17-27 km, as expected from Fig. 3. The strongest correlations occur near zero time lag, but the lag correlations are skewed towards positive lags, which is a signature of ozone leading temperature anomalies by a small amount, similar to the annual cycle in Fig. 2b.

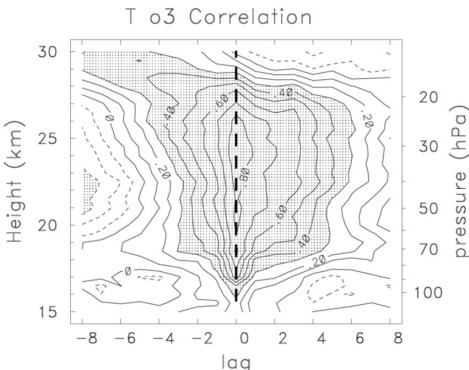

Figure 4. Correlation of deseasonalized SHADOZ ozone and temperature time series as a function of height and time lag (in months).
        Positive lag denotes ozone leading temperature.

        A scatter plot of the SHADOZ ozone-temperature deseasonalized anomalies at 24 km over 1998-2016 is shown in

Fig. 5, highlighting the strong observed correlation. The slope of the (O3/T) variations is near 0.14 (ppmv/K). This slope

changes as a function of altitude (as shown below), and this is one of the quantities that we aim to understand from a simple

perspective.

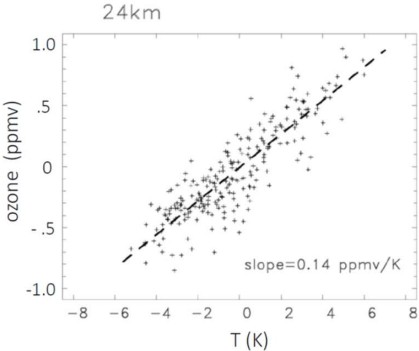

Figure 5. Scatter plot of deseasonalized ozone and temperature anomalies from SHADOZ data at 24 km (data as in Fig. 3).

**4.2 Satellite observations**

        We use MLS and GPS satellite data to quantify ozone-temperature correlations over a continuous range of time

scales from days to over a decade. Time series of zonal mean MLS ozone and GPS tempertures over the equator (10° N-S) at

24 km (31 hPa for MLS) are shown in Fig. 6, for pentad averages covering September 2004 – March 2018.  Visual





inspection of Fig. 6 shows a clear signature of the QBO (as in Fig. 3), and strong correlations of ozone and temperature

across all scales of variability.

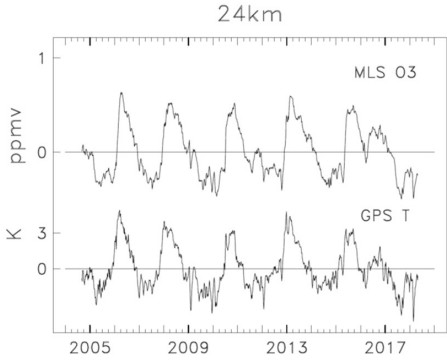

Figure 6. Time series of zonal average MLS ozone and GPS temperature at 24 km, for averages over 10° N-S.

Power spectra for ozone and temperature at 24 km are shown in Fig. 7a. The spectra for both quantities show most

power at low frequencies, with peaks linked to the QBO, annual and semi-annual cycles. At altitudes below 24 km the
annual cycle is more pronounced, while above 24 km the semi-annual cycle is larger (e.g. Fig. 2). Power decreases
systematically at periods shorter than semi-annual for both ozone and temperature in Fig. 7a. Ozone-temperature coherence
squared (coh$^2$) at 24 km is shown in Fig. 7b, highlighting significant values over nearly the entire range of periods longer
than ~20 days. There is a relative minimum in coh$^2$ near the semi-annual cycle, and this could possibly be related to the

semi-annual variation in low-latitude ozone photochemical production noted above, which adds additional ozone variability
that is less coherent with temperature. The reason for the lack of coherence at the shortest resolved time scales (<20 days) is
unknown, but could be related to very low power in both data sets (Fig. 7a) and poorer temporal resolution based on pentad
data. There is a relatively small phase difference between ozone and temperature over all frequencies, as shown below.
Similar behavior is found for ozone-temperature coh$^2$ and phase for all altitudes over 17-27 km. Above 29 km there is a

strong coh$^2$ maximum for the semi-annual oscillation (~180 days period), where ozone and temperature are approximately
out of phase (not shown). This behavior is due to the transition to photochemical control of ozone and the impact of
temperature on the $O_x$ loss rate (Brasseur and Solomon, 2005).

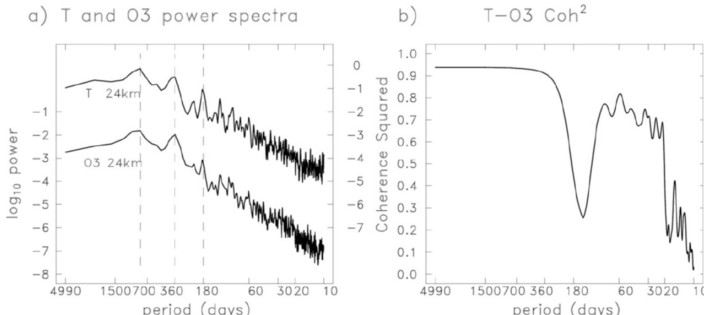

Figure 7. (a) Power spectra for GPS temperature and MLS ozone at 24 km as a function of wave period. Logarithmic power units are ($K^2$ –
right axis) and ($ppmv^2$ – left axis). The vertical dashed lines identify peaks in the spectra associated with the QBO, annual and
semi-annual cycles. b) Coherence squared between temperature and ozone at 24 km as a function of wave period.

## 5 Comparisons with idealized model calculations

We next compare frequency-dependent ($X_\sigma/T_\sigma$) amplitude and phase between observations and results from the
idealized model calculations (Eqns. 9a-b). Figure 8a compares observed and modeled 24 km ($X_\sigma/T_\sigma$) amplitude as a function
of frequency. Observations are based on the MLS ozone/GPS temperature results, where ($X_\sigma/T_\sigma$) is calculated from the
respective harmonic coefficients and the ratio is smoothed in frequency; model results are shown with and without including
ozone damping effects, which has relatively small influence. Additionally, Fig. 8a includes the ($X_\sigma/T_\sigma$) ratio estimated from
deseasonalized SHADOZ anomalies (from Fig. 5) which are mainly associated with the QBO (~28 months period). The
observed ($X_\sigma/T_\sigma$) ratio shows a systematic change over the frequency range, with approximately a factor of two increase in
the ratio for low frequencies (periods > 150 days) compared to high frequencies. There is a relative maximum in the ($X_\sigma/T_\sigma$)
ratio near the semi-annual period; the cause of this feature is not understood but might be related to the semi-annual ozone
photochemical production term over the equator discussed above, which is not included in the model calculations. The
idealized model results show a similar ($X_\sigma/T_\sigma$) frequency dependence, although with substantial disagreement on the shape
of the transition region between semi-annual and interannual (QBO) periods, with a much smoother transition in the model.
The systematic change with frequency corresponds to the change from ozone-temperature coupling via transport (high
frequency) to radiative balance (low frequency). Including the ozone damping ($\delta$) improves the agreement at low
frequencies. The observed and modeled ($X_\sigma/T_\sigma$) phase relationship as a function of frequency is shown in Fig. 8b, showing
approximately in-phase behavior across all frequencies in both cases. The model ($X_\sigma/T_\sigma$) phase is slightly negative (as
expected from Section 3), while the observed values are near zero or slightly positive. Similar behavior to Fig. 8 is found in
the satellite data for all altitudes over 19-27 km.





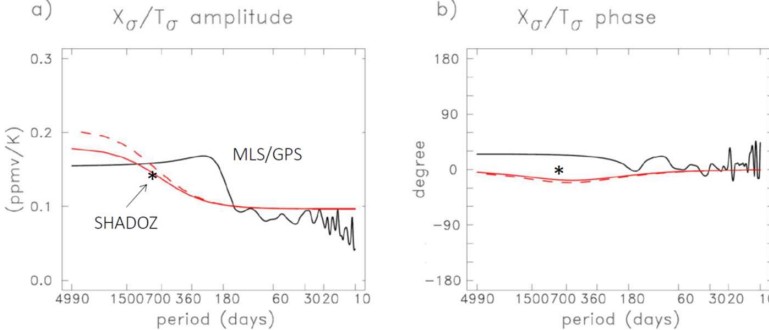

Figure 8. (a) $(X_\sigma/T_\sigma)$ ratio at 24 km as a function of wave period. Black line shows observational results from MLS/GPS satellite data, and red lines show idealized model calculations (solid red line includes ozone damping, and dashed red line is for δ=0). The $(X_\sigma/T_\sigma)$ ratio derived from deseasonalized SHADOZ data at 24 km (Fig. 5) is also shown, which is mainly associated with the QBO (~28 months period). (b) corresponding $(X_\sigma/T_\sigma)$ phase differences.


Vertical profiles of observed and modeled $(X_\sigma/T_\sigma)$ amplitude are shown in Fig. 9 for three different frequency bands, corresponding to 'fast' frequencies (30-60 days period, Fig. 9a), annual cycle (Fig. 9b) and QBO (Fig. 9c). In addition to observations from the MLS/GPS data, we include the corresponding ratios calculated from SHADOZ ozone and

temperature data for the annual cycle (e.g. Fig. 2a), calculated as the ratio of the respective ozone and temperature maximum-minimum values over the annual cycle, for altitudes 17-23 km where the annual cycle is distinct in the data. Figure 9c includes SHADOZ results for deseasonalized anomalies, which are mainly linked to the QBO, and derived from regression as in Fig. 5. These SHADOZ results in Figs. 9b-c agree well with the corresponding estimates from MLS/GPS satellite data. The fast frequencies (Fig. 9a) are governed by vertical transport with a $(X_\sigma/T_\sigma)$ vertical profile close to $(X_z/S)$

(Fig. 1c), and the model shows a good fit to the observed vertical structure, at least up to ~27 km. The annual cycle (Fig. 9b) is close to the cross-over between high- and low-frequency behavior, and the model again shows approximate agreement to observations over altitudes where the annual cycle is large (~17-23 km). This agreement helps confirm the interpretation that the annual cycles in tropical stratospheric ozone and temperature (e.g. Fig. 2) can be interpreted as coupled responses to the annual cycle in tropical upwelling, with ozone feeding back on temperature. There is poorer agreement in Fig. 9b for the

annual cycle above 23 km, but over these altitudes the actual variability has mainly a semi-annual component. For the lower frequency QBO variations (Fig. 9c) the idealized model shows good agreement with the $(X_\sigma/T_\sigma)$ amplitude from both the satellite data and SHADOZ with a peak near 27 km, and note that including ozone damping (δ) slightly improves the agreement above ~25 km. Our conclusions from the comparisons is that the idealized model can quantitatively explain the observed $(X_\sigma/T_\sigma)$ amplitude and phase relationships in the tropical lower stratosphere, including their dependence on

frequency and altitude.



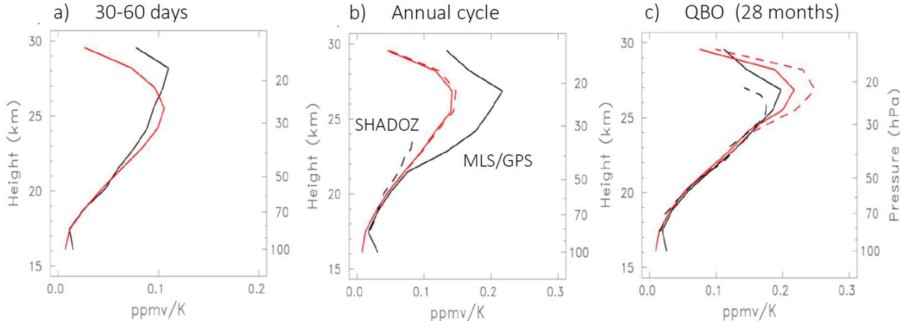

Figure 9. Vertical profiles of $(X_\sigma/T_\sigma)$ amplitude for selected frequency bands. (a) 30-60 day periods, (b) annual cycle (12 months period), (c) QBO period (~28 months). The solid black lines are results from MLS/GPS satellite data, and red lines show idealized model calculations (solid red line includes ozone damping, and dashed red line is for δ=0). Additionally, results from SHADOZ ozone and temperature data are included as dashed black lines in (b-c), as described in text. Note the horizontal axes change among the three panels.

## 6 Summary and discussion

Observations show strong correlations between ozone and temperature in the tropical lower stratosphere, and calculations show that the ozone radiative feedbacks significantly enhance temperatures, e.g. by ~30% for the annual cycle (e.g. Ming et al, 2017). This ozone feedback significantly enhances thermal variability in global model simulations (Yook et al, 2020). The goals of this work include providing an update of observational evidence for O3-T coupling and simplified understanding based on idealized zonal mean theory. The excellent long-term tropical ozonesonde measurements from SHADOZ demonstrate approximate in-phase O3-T correlations for the annual cycle (Fig. 2) and for interannual anomalies (Figs. 3-5), which are dominated by the QBO. Long-term continuous satellite measurements from zonal average MLS and GPS data agree well with these results for annual and interannual variations, and furthermore demonstrate strong O3-T coherence for faster sub-seasonal variability (Figs. 6-7b). This coherent behavior is observed throughout the lower to middle stratosphere, ~17-27 km, with O3-T anomalies approximately in phase over all altitudes. A key result is that the observed (O3/T) ratio changes as a function of frequency, with approximately twice the ratio for low frequencies (annual cycle and longer) compared to faster variability (Fig. 8a). The (O3/T) ratio also depends on altitude, with larger ratios in the middle stratosphere (Fig. 9). These are the key observational characteristics of O3-T coupling that we seek to explain.

We compare observations to results from idealized zonal mean theory, assuming vertical advection from the upward Brewer-Dobson circulation controls thermal balances and ozone transport, i.e. neglecting mean meridional advection and eddy transport terms. This is a reasonable approximation in the tropical lower stratosphere (Abalos et al, 2013), although eddy transport from monsoon circulations makes important contributions to ozone tendencies during boreal summer at and



below ~18 km (Konopka et al, 2009, 2010; Stolarski et al, 2014). Thermodynamic balance includes linear radiative damping ($\alpha$) and ozone feedback ($\beta$) terms, and the coupled equations (including linear ozone damping $\delta$) can be solved analytically to calculate the (O3/T) ratio as a function of frequency and altitude, dependent on model parameters $\alpha$, $\beta$ and $\delta$ and the ratio of background gradients expressed as ($X_z/S$). In general, ozone damping is a minor influence over most of the domain because of the long ozone lifetimes. The model balances highlight two time scales for O3-T coupling, including 'fast'

variability where the (O3/T) ratio is determined by the background vertical gradients ($X_z/S$) and 'slow' time scales determined by radiative balance ($\alpha/\beta$). The idealized model shows a functional frequency dependence for the (O3/T) ratio similar to observations (Fig. 8a), although there is disagreement in the transition region where the observations have a more rapid (O3/T) transition with a relative maximum near the semiannual period. This detail is not well understood, but could be influenced by a semiannual ozone photochemical production term in the equatorial region related to solar inclination (Abalos

et al, 2013) that is not included in our model, along with neglected eddy transport effects, especially near the tropopause. This semi-annual ozone production may also explain the relative minimum in O3-T coherence squared near this frequency found in Fig. 7b. The vertical profiles of (O3/T) ratio agree well with the observations for both fast (Fig. 9a) and slow (Figs. 9b-c) time scales, enhancing confidence in a simple understanding. We note that the frequency-dependent O3-T behavior shown here is consistent with the results of Charlesworth et al (2019), which indicate larger ozone radiative impacts on

tropopause temperatures for low frequencies.

It is worthwhile to appreciate the limitations associated with the idealized model calculations, especially uncertainties related to the parameters $\alpha$ and $\beta$, which control the low frequency model behavior. While Hitchcock et al (2010) show that linear regression on temperature captures ~80% of the variance in modeled radiative heating rates in the tropical lower stratosphere, the broad spectrum of vertical scales in this region can introduce additional uncertainties in

estimating $\alpha$. Our calculations of ozone heating via the $\beta$ term in Eq. 3 neglects the effects of non-local ozone changes, which will also depend in detail on the vertical scale of perturbations. In spite of these caveats, the overall agreement between model and observations demonstrates that the idealized zonal mean theory (quantifying coupled O3-T response to variations in the Brewer-Dobson circulation) is a valid perspective to understand the strong O3-T coupling in the real atmosphere.


**Acknowledgements**

The National Center for Atmospheric Research is sponsored by the U.S. National Science Foundation. This work has been partially supported by the COSMIC NSF-NASA Cooperative Agreement under Grant 1522830, and by the NASA Aura Science Team under Grant 80NSSC20K0928. AM would like to acknowledge support from the Leverhulme Trust as

an early career fellow. We thank Marta Abalos and Rolando Garcia for discussions and comments that significantly improved the manuscript.



**Data Availability**

SHADOZ data were obtained from the SHADOZ website https://tropo.gsfc.nasa.gov/shadoz/. MLS ozone data were
obtained from https://mls.jpl.nasa.gov/index-eos-mls.php, and GPS temperatures were obtained from the COSMIC Data
Analysis and Archive Center (CDACC) website https://cdaac-www.cosmic.ucar.edu/.

**Author contributions**

The study was conceived by WJR, and data analysis was performed by FW. AM and PH provided input for the idealized
model calculations and contributed to interpretation of results. The paper was written by WJR, with editing from AM and
PH.

**Competing interests**

The authors declare that there are no competing interests.

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
