# Peer review of "A simple model of ozone-temperature coupling in the tropical lower stratosphere"

_Atmospheric Chemistry and Physics, 2021_

## Author Comment (AC1)

We thank the reviewers for the positive and constructive reviews. We include below detailed responses to the individual comments. One important change to the paper in response to the reviewer comments is that we now focus on the (T/O3) ratios rather than the (O3/T) ratios, which allows direct evaluation of the ozone feedback onto temperature. This is a straightforward change in our calculations and presentation of results, with corresponding changes to Figs. 1c, 5, 8 and 9. We have also revisited our calculations of T-O3 phase difference from observations and have corrected a previous sign error, resulting in better agreement with the idealized model (new Fig. 8b).

**Reviewer 1**

This paper by Randel et al. investigates the coupling between ozone and temperature variability in the tropical lower stratosphere. Related information is extracted from different observational datasets, including balloon and satellite data, and compared to results from an idealized model based mainly on zonal mean vertical advection and radiation. A particular focus is laid on the O3/T ratio which is found to change as a function of frequency and altitude. The idealized model results compare qualitatively well with the observation-based results showing that the common variability in tropical ozone and temperatures can indeed be interpreted as coupled response to variability in tropical upwelling, including ozone radiative feedback. Furthermore, the idealized model shows that for fast frequency variability the O3/T ratio is controlled by transport whereas for slow variations it is controlled by radiative processes.

I like the approach of comparing observational data to an idealized model to shed new light on the ozone-temperature coupling in the tropical lower stratosphere. The subject of the paper will be of interest to a wide readership from the atmospheric science and climate communities. Overall, the paper is very well and fluently written and the results are presented in a clear and concise way. There are no major concerns regarding publication, only a few minor and specific comments which came into my mind while reading and which perhaps could further improve the paper.

Thanks for the positive comments.

Minor comments:

1) Feedback effect of ozone on temperature

One main question I had when starting reading was how important the radiative feedback from ozone on temperatures would be in comparison to the response to upwelling. After reading I'm still unsure of what I do learn from this paper quantitatively in that respect. It is clear that the idealized model including the feedback term does a fairly good job. But, how important is the feedback term at all (i.e. the beta term in the thermodynamic equation)? Wouldn't it make sense to include results from the idealized model without the feedback term (beta=0) and quantify the feedback effect from the difference to the full model results?

This is an excellent comment that made us rethink our calculations. We have now reformulated the calculations in terms of the (T/O3) ratios instead of (O3/T), and now include model results for beta=0 that allow direct estimation of the ozone feedback on temperature as a function of frequency (and altitude). Accordingly, there are new Figs. 8-9, and we have added new discussions of quantifying ozone feedbacks on temperature (that are largest for low frequency variability) in Sections 3.1, 5 and 6.

2) Agreement between observations and idealized model

I'm unsure whether the agreement between the idealized model and observationally-based results is really well from a "quantitative" point of view, as stated a few times in the paper (e.g., L300, L308, L347). At other places it is said that the agreement is "approximate" (e.g., L19), a wording I would prefer given the clear differences e.g. in Figs. 8/9. I agree that the frequency and altitude dependence is qualitatively well reproduced by the simple model, but clear quantitative differences remain. In particular, the phases in Fig. 8b are even opposite over a wide spectrum range. I suggest to carefully check and adjust the wording in this regard before publication.

The new calculations using (T/O3) ratios involved some rewriting of the text, and we have tried to be conservative in evaluating the agreement between model and observations. We have also revisited our calculations of T-O3 phase difference and discovered a sign error in our previous observational results, which is corrected in the new Fig. 8b. (also, please note the T-O3 phases are opposite to the O3-T phases in the previous version of the paper).

Specific comments:

L42: It could be mentioned here for completeness that horizontal transport also plays a role in the tropical ozone budget, but mainly at lower altitudes around the tropopause (e.g., Konopka et al., 2010; Abalos et al., 2013b), although this is mentioned already later.

This detail is already mentioned several times in the manuscript and doesn't fit well in the flow of the discussion at this point, so we have not made any changes.

L90ff: I would suggest to mention once at beginning of this section that all variables in the formulas below are zonal mean quantities, but that no overbar is used in the notation.

Done.

L118: Here, it would already be good to mention that meridional advection and eddy transport play also a role, but that it has been shown that this effect becomes very small above 80 hPa (e.g., Abalos et al., 2013b) and therefore can indeed be neglected to a good approximation here. At levels closer to the tropopause (up to about 18km), neglecting horizontal transport will cause some bias to the presented results. Similar statements are made later in L150ff - but I think it would be good to say that already here.

We agree that these neglected terms can be important below ~80 hPa, and this is clearly stated in this section.

L124: How large can the effect of variability in the background ozone gradient be? In other words, how good is the approximation of constant background gradient?

It is already noted below in the paper (~line 143) that variations of ~10% in stability and ozone vertical gradient are typical.

L196ff and Fig. 2a: As the seasonal cycle is shown at 18km and horizontal transport can have an effect on the tropical ozone budget below 19km, as even said a sentence later, it is not clear to me to what degree the statement "correlated ozone-temperature behaviour is mainly a response to the annual cycle in upwelling" holds. Maybe it would be better to show the results for 19km in Fig. 2a, or being more careful with wording.

We have changed Fig. 2 to 19 km.

L243: Perhaps say that 24km is considered here as it is the level with strongest correlation (see Fig. 4) - but that the conclusions also hold for other levels (at least, that's what I guess...).

We have added a statement that quantitatively similar results are found over ~17-27 km.

L272ff and Fig. 8: Why is only the QBO amplitude shown for SHADOZ? It would be great to include results also for other frequencies (e.g., seasonal cycle amplitude is also included in Fig. 9).

The QBO is highlighted because that is the dominant scale of variability, e.g. Fig. 3, in addition to the annual cycle over ~19-23 km. The (T/O3) ratios for the QBO time scale can be calculated directly from regressions (e.g. Fig. 5), which is why we include that estimate in Fig. 8. We have chosen not to perform separate cross-spectrum analysis for the monthly SHADOZ data to keep the analyses relatively simple.

L280: Why does the change from one regime to the other (transport vs. radiation controlled) occurs around a period of about 200 days? Can this be understood from the idealized model?

Good question. We have added some discussion in the next-to-last paragraph of Section 3.1 (line 162+) regarding the frequency transition between the low- and high-frequency regimes.

L285: Any ideas why there is this discrepancy in phase between the model and observational results? Doesn't this mean that causality is flipped, hence a more substantial difference? (Which is related to my minor comment 2). Including some more discussion here would be helpful for interpretation.

As noted above, we discovered a sign error in our previous calculations of the observed (T-O3) phase differences. This is now corrected, and the model comparisons show much better agreement (new Fig. 8b).

L299 and Fig. 9a: How about including the ratio of background gradients in Fig. 9a (the profile from Fig. 1c) to ease comparison?

Good idea, but the results lie directly below the 'full' model calculation. This is now mentioned in the text.

L332: After (Abalos et al., 2013) add "above the TTL".

Done.

Fig. 8: What is the x-axis scaling? Maybe mention explicitly in the caption that it is not linear.

We have now explicitly explained the x-axis of the spectral plots, near line 255.

Fig. 9: Perhaps better to have the same x-axis range for all three subfigures.

Figure 9 is now changed, but we use the same x-axis range for all panels.

---

## Author Comment (AC2)

We thank the reviewers for the positive and constructive reviews. We include below detailed responses to the individual comments. One important change to the paper in response to the reviewer comments is that we now focus on the (T/O3) ratios rather than the (O3/T) ratios, which allows direct evaluation of the ozone feedback onto temperature. This is a straightforward change in our calculations and presentation of results, with corresponding changes to Figs. 1c, 5, 8 and 9. We have also revisited our calculations of T-O3 phase difference from observations and have corrected a previous sign error, resulting in better agreement with the idealized model (new Fig. 8b).

**Reviewer 2**

Review of Randel et al. "A Simple model of ozone-temperature coupling in the tropical lower stratosphere"

This study examines the strong correlations between ozone and temperature variations in the tropical lower stratosphere over a range of time scales using both ground-based ozone/temp. from SHADOZ balloon measurements and satellite based measures combining MLS ozone with GPSRO temperature measurements. Noting the strong in-phase relationship largely driven by circulation variations. The authors work through the process of developing a simple model for this relationship and discuss the timescales, locations, and feedbacks that are operating. The paper is well organized and clearly written and would be a very welcome addition and certainly of interest to the readership. I think the paper could be published as is but I have a couple very minor comments that the authors might consider.

Thanks for the positive comments.

There is discussion (page 8) of the photochemical control becoming dominant above 27 km, I would imagine that there is still some significant contribution at 24 km since it is in the transition region, how much of the O3/T relationship is begin impacted by photochemistry (might be opposite direction, i.e. increased T decreased O3) at that level.

The ozone photochemical lifetime at 24 km is >100 days, and photochemical changes are small compared to dynamic variations. Furthermore, the O3/T results at 24 km agree quantitatively with the model with no photochemistry (aside from ozone damping). From Fig. 9, the idealized model (transport only) works quantitatively well up to 27 km.

Lines 275-277 and figure 8 related to the unexplained difference in the semi-annual - annual (X sigma/T sigma) magnitude, this is referenced to 24 km, although lines 284-285 mention similar behavior over a range of altitudes. Does the vertical behavior provide any additional information? Is there any way to look at SHADOZ beyond just the QBO periodicity, do the SHADOZ profiles have to be deseasonalized.

We've looked carefully at the vertical structure of the observed and modeled spectra, but the behavior is similar at other levels and we decided to focus on one level. We examined various

vertical profile diagnostics but could not find anything additional to add. The annual cycle and QBO signals are the dominate variability in SHADOZ data, and we focus on those signals.

For Figure 9 why does the SHADOZ amplitudes only go to 24 km and in this figure there is SHADOZ data for annual cycle so is it not deseasonalized for this figure. From the levels that are shown the amplitude of the annual response from SHADOZ is a bit smaller than the model with MLS/GPSRO showing much larger amplitudes and a growing difference above 22 km, any thoughts if the difference in vertical resolution between the relatively lower resolution ozone and higher vertical resolution GPSRO temperatures could be playing a role or something else.

The large annual cycle in ozone and temperature occur over altitudes ~16-22 km in the lower stratosphere, and variability is primarily semi-annual above ~24 km.  Hence we focus the annual cycle comparisons over altitudes up to 23 km in Fig. 9b, and do not include SHADOZ or MLS/GPS above that level (changed from the previous version). There are some small differences between the SHADOZ and MLS/GPS annual cycle results (due to vertical resolution and very different sampling, among other things), but overall there is quite reasonable agreement with the model calculations up to 23 km.

Have you considered looking at what the MLS Ozone/MLS Temperature amplitudes would produce?

The MLS temperatures agree well with GPS data, but have somewhat lower vertical resolution. We have not redone the calculations using MLS temperatures.